

# A retrospective study on beta-blocker use and outcomes in hematopoietic stem cell transplant patients

Matthew A. Bergens[1], John T. Bokman[1], Ernaya J. Johnson[2], Matthew L. Braun[3], Yan Li[4], Amy T. Bush[2], Lauren Hill[2], Jolien Van Opstal[5], Alessandro Racioppi[1], Rebecca Fan[6], Sejal Kaushik[6], Edwin Alyea[2], Nelson Chao[2], Taewoong Choi[2], Cristina Gasparetto[2], Mitchell Horwitz[2], Richard Lopez[2], Sendhilnathan Ramalingam[2], Keith Sullivan[2], Paul Wischmeyer[7] and Anthony D. Sung[3]

[1] School of Medicine, Duke University, Durham, NC, United States of America
[2] Division of Hematologic Malignancies and Cellular Therapy, Duke University, Durham, NC, United States of America
[3] Division of Hematologic Malignancies and Cellular Therapy, University of Kansas Medical Center, Westwood, KS, United States of America
[4] Duke Cancer Institute Biostatistics Shared Resource, Duke University, Durham, NC, United States of America
[5] Faculty of Medicine, Katholieke Universiteit Leuven, Leuven, Belgium
[6] Duke University, Durham, NC, United States of America
[7] Departments of Anesthesiology and Surgery, Duke University, Durham, NC, United States of America

Corresponding author
Anthony D. Sung, asung2@kumc.edu

## ABSTRACT

Recent studies have linked beta-blocker (BB) use in critical care settings with improved survival outcomes, potentially due to beta-adrenergic receptor ($\beta$-AR) blockade and associated anti-inflammatory effects. Given the immune system's role in the development of graft-*versus*-host disease (GVHD)—a major complication of allogeneic hematopoietic stem cell transplant (allo-HCT)—we conducted a single-center retrospective review to assess the impact of BB use on acute GVHD (aGVHD) and other survival outcomes in patients undergoing their first allo-HCT. We analyzed 10 years of data (January 2010 to May 2020), including 105 patients who received a BB for more than four days before and after HCT, and 669 control patients who did not receive a BB. Patients on BBs had a lower incidence of aGVHD (55.2% *vs.* 65.8%, $p = 0.036$); however, this difference was not statistically significant in multivariate analysis ($p = 0.150$). When stratified by BB mechanism, outcomes varied: non-selective BBs were associated with lower post-HCT weight ($p = 0.034$), and vasodilating BBs showed a borderline reduction in length of stay (LOS) ($p = 0.054$). While our findings confirm the pharmacological safety of BBs in this population, they do not support their routine use for modifying allo-HCT outcomes. Future prospective studies with larger cohorts are needed to further explore the role of BBs in peri-HCT management and to clarify their clinical implications and therapeutic potential.

## INTRODUCTION

Allogeneic hematopoietic stem cell transplant (allo-HCT) is an effective treatment for hematological malignances, but its success is often complicated by graft-*versus*-host disease (GVHD). GVHD arises from a dysregulated immune response, where donor T cells attack host tissues. It is intricately connected to a cascade of pro-inflammatory signals through both cytokine release and immune cell activation, which leads to sustained tissue damage and inflammation (*Ferrara, Cooke & Teshima, 2003*; *Holler, 2002*; *Piper & Drobyski, 2019*; *Jankovic et al., 2013*). Approximately 30–60% of allo-HCT recipients develop acute GVHD (aGVHD) and the Center for International Blood and Marrow Transplant Research (CIBMTR) estimated that 12–14% of all deaths among HLA-matched allo-HCT recipients from 2018–2020 were due to GVHD (*Bolon et al., 2022*). In severe cases, patients with a grade III–IV acute GVHD (aGVHD) suffered a 29% transplant-related mortality rate (*Jagasia et al., 2012*; *Khoury et al., 2017*). Beyond its impact on mortality, aGVHD is also associated with increased hospital length of stay (LOS) and overall healthcare costs (*Yu et al., 2019*). Current first-line treatment for GVHD consists of immunosuppression with corticosteroids; however, this approach is only effective in 55–65% of cases, particularly with patients with severe GVHD (*Flowers & Martin, 2015*; *Axt et al., 2019*; *Calmettes et al., 2015*). Given the significant impact GVHD has on patients and the lack of a more reliable treatment, there is a need for improved prophylaxis to mitigate GVHD severity or incidence.

Beta-blockers (BB) are pharmacologically safe medications commonly used clinically for cardioprotection by modulating the beta-adrenergic receptor (b-AR) (*Gorre & Vandekerckhove, 2010*; *Oliver, Mayor Jr & D'Ocon, 2019*). However, recent research has highlighted their immunomodulatory effects in various clinical settings, including critical illness (*Wilson et al., 2013*; *Morelli et al., 2013*), burn injury (*Herndon et al., 2001*; *Kopel et al., 2021*) and cancer (*Grytli et al., 2014*; *Hochberg, Cairo & Friedman, 2014*; *Jansen et al., 2014*; *Wang et al., 2013*; *Hwa et al., 2017*). BB use in critically ill patients is associated with improved outcomes due to reduced systemic inflammation and attenuation of excessive immune activation (*Wilson et al., 2013*; *Morelli et al., 2013*). In oncology, BBs are associated with improved outcomes in various solid tumors and hematologic malignancies, including overall survival and disease-specific mortality, potentially due to mechanisms related to immune surveillance and inflammatory signaling (*Grytli et al., 2014*; *Hochberg, Cairo & Friedman, 2014*; *Jansen et al., 2014*; *Wang et al., 2013*; *Hwa et al., 2017*). These findings suggest that BBs may play a role in regulating inflammatory responses, which are central to GVHD pathogenesis.

BBs have previously been investigated in the setting of allo-HCT. One study on propranolol demonstrated a potential survival benefit, as well as underscored its feasibility and tolerability in allo-HCT patients (*Knight et al., 2018*). Another phase two randomized controlled trial found that propranolol inhibited stress-related pathways, suggesting potential benefits in reducing relapse and improving disease-free survival (*Knight et al., 2020*). Different b-AR receptor subtypes may have distinct role in modulating immune responses. In previous studies, B1-selective BBs have shown protection against

catecholamine-induced injury in critical illness (*Kuo et al., 2021*; *Heliste et al., 2022*), while b2-AR has been implicated in enhancing the graft-*versus*-tumor (GVT) effect (*Mohammadpour et al., 2018*), and b3-AR is involved in modulating oxidative stress in hematopoetic stem cell redox homeostasis (*Pasha, Calvani & Favre, 2021*).

Since b-AR signaling impacts a wide variety of immune responses, including pro-inflammatory pathways (*Ferrara, Cooke & Teshima, 2003*; *Holler, 2002*; *Piper & Drobyski, 2019*; *Jankovic et al., 2013*; *Powell et al., 2013*), hematopoesis (*Maestroni, 2020*; *Méndez-Ferrer, Battista & Frenette, 2010*), and hematopoietic reconstitution after allo-HCT (*Wang & Cao, 2019*), BBs may have a beneficial role in allo-HCT. Blockade of b-AR signaling and the associated pro-inflammatory pathway could be a promising strategy to mitigate GVHD and improve post-HCT outcomes. Thus, we hypothesize that BB use prior to allo-HCT may be associated with decreased GVHD and improved survival outcomes.

## METHODS

### Study design

This is a single center retrospective study that investigated a potential relationship between BB use and survival outcomes during allo-HCT. Patient data was pulled from the electronic health records (EHR) and then reviewed manually to ensure accuracy. A retrospective study design was chosen due to the availability of patient data in the Duke Adult Bone Marrow Transplant (ABMT) database, allowing for a sufficiently large sample size. This study allowed us to analyze real-world data without the time and resource constraints of a prospective trial. However, we recognize the limitations of a retrospective approach, such as potential selection bias, reliance on accurate documentation in medical records, and the inability to establish causation. Approval for exemption was obtained from the Duke University institutional review board (Pro00103818).

### Patient population and data collection

All patients who received their first allo-HCT between January 2010 and May 2020 at the Duke ABMT clinic were included in this retrospective analysis. This timeframe was selected to maximize sample size while attempting to minimize variation in treatment regimen and inconsistencies from older charts. Demographic data including age, gender, transplant type, conditioning regimen, underlying hematological malignancy, and Hematopoietic Cell Transplantation-specific Comorbidity Index (HCT-CI) score before transplantation, were collected from the Duke ABMT database. Pre-transplant data was collected at D-10 and post-transplant data was collected at D+90. Outcomes such as GVHD occurrence and grade, length of stay (LOS), non-relapse mortality (NRM), relapse occurrence, overall survival (OS), and cause of death, were abstracted from the Duke ABMT Database.

All patients were chart reviewed to confirm their exposure to BB as well as record the specific BB administered. Patients were included in the BB group if they had exposure to a BB both before and after transplantation (peri-HCT). Due to limitations in record availability (2010–2020), precise duration of BB use could not always be determined. Additionally, the relationship between dose, plasma concentration, and clinical effect is variable among both individuals and different BBs, thus there is no universally accepted

dose or duration that defines clinically significant beta-blocker exposure (*Kendall, 1997*; *Heidenreich et al., 2022*). Consequently, we selected an inclusion criterion requiring at least four consecutive days of BB use both before and after HCT, consistent with previous studies (*Wijeysundera et al., 2014*). This threshold was chosen to focus on patients with sustained BB exposure likely to exert meaningful physiologic impact, while excluding patients who initiated but quickly discontinued BBs due to intolerance or other medical concerns. Patients who had received a BB were then subdivided by the common clinical subdivisions of BB mechanism (selective BBs, non-selective BBs, and vasodilating BBs) to explore potential differences in outcomes based on pharmacologic properties (*Oliver, Mayor Jr & D'Ocon, 2019*).

## Statistical analysis

The primary outcome endpoints included both aGVHD occurrence-rate and grade. Other secondary outcomes include chronic GVHD (cGVHD) occurrence-rate and grade and OS, NRM, relapse occurence, and LOS. For OS, the event is defined as all-cause death, and censured at last follow up. For NRM, the event is defined as date of relapse, and censured at last follow up. LOS is defined as number of days between transplant date and discharge from peri-HCT care, back to the patient's local oncologist. Comparisons of patient characteristics were performed using Chi-squared test or Fisher's exact test for categorical variables, and the analysis of variance or Wilcoxon Rank Sum test for continuous variables, respectively. The survival analyses were performed using the Kaplan–Meier method. The log-rank test was applied to detect overall group differences in outcome endpoints. Multivariate analysis (MVA) with the Cox proportional hazard model was used to evaluate the association of covariates and the aGVHD-free rates. The response variable of interest for the MVA was the occurrence of aGVHD, and the covariates include the variables that had significant effects in univariate analyses.

## RESULTS

Demographics were compared between patients who were on a BB before and after allo-HCT (Yes BB) ($n = 105$) and those who were not (No BB) ($n = 669$). The Yes BB group had a greater proportion of participants with an HCT-CI of 4 or greater ($p = 0.010$). However, there were no significant differences between the two cohorts by race, sex, median age, conditioning regimen, or donor type (all $p > 0.05$) (Table 1).

Patients in the Yes BB group had a significantly lower incidence of acute GVHD (55.2% *vs.* 65.8%, $p = 0.036$) (Table 2). Median albumin levels were lower in the Yes BB group at D-10 ($p = 0.042$) and D+90 ($p = 0.014$) than in the No BB group; however, the change in Albumin between D-10 and D+90 was not significant ($p = 0.101$). There were no observed differences in cGVHD occurrence or grade or other survival outcomes such as OS, NRM, relapse occurrence, and LOS (all $p > 0.05$) (Table 2, Figs. S1A–S1E). We performed multivariate analysis to further examine associations between BB use and other co-variates on aGVHD (Table 3). We included covariates that were associated with aGVHD on univariate analysis, as well as age at transplant and type of GVHD prophylaxis. Albumin
**Table 1  Demographics of patients who were not on a BB (No BB) or who were on a BB (Yes BB) peri-transplant (> 4 days both before and after) to allo-HCT.**

| Parameter Statistic | All patients N = 774 (100%) | No BB N = 669 (86%) | Yes BB N = 105 (14%) | P-Value |
|---|---|---|---|---|
| Age (median, IQR) | 53.00 (42.00–61.00) | 53.00 (42.00–61.00) | 55.00 (46.00–63.00) | 0.065 |
| Gender, n (%) | | | | 0.651 |
|     Female | 303 (39.15) | 264 (39.46) | 39 (37.14) | |
|     Male | 471 (60.85) | 405 (60.54) | 66 (62.86) | |
| Race, n (%) | | | | 0.307 |
|     White | 634 (81.91) | 553 (82.66) | 81 (77.14) | |
|     Black | 117 (15.12) | 98 (14.65) | 19 (18.10) | |
|     Other/Unknown | 23 (2.97) | 18 (2.69) | 5 (4.76) | |
| Ethnicity, n (%) | | | | 0.521 |
|     Non-Hispanic | 723 (93.41) | 625 (93.42) | 98 (93.33) | |
|     Hispanic | 13 (1.68) | 10 (1.49) | 3 (2.86) | |
|     Unknown/unreported | 38 (4.91) | 34 (5.08) | 4 (3.81) | |
| Transplant Diagnosis, n (%) | | | | 0.835 |
|     Acute Leukemia | 362 (46.77) | 314 (46.94) | 48 (45.71) | |
|     Chronic Leukemia | 59 (7.62) | 50 (7.47) | 9 (8.57) | |
|     Lymphoma | 125 (16.15) | 111 (16.59) | 14 (13.33) | |
|     MDS/MPN | 176 (22.74) | 151 (22.57) | 25 (23.81) | |
|     Other | 52 (6.72) | 43 (6.43) | 9 (8.57) | |
| Graft, n (%) | | | | 0.520 |
|     Allo matched related | 233 (30.10) | 203 (30.34) | 30 (28.57) | |
|     Allo matched Unrelated | 328 (42.38) | 279 (41.70) | 49 (46.67) | |
|     Cord blood | 119 (15.37) | 106 (15.84) | 13 (12.38) | |
|     Haploidentical | 73 (9.43) | 61 (9.12) | 12 (11.43) | |
|     Other | 21 (2.71) | 20 (2.99) | 1 (0.95) | |
| Conditioning regimen, n (%) | | | | 0.580 |
|     Myeloablative | 497 (64.21) | 434 (64.87) | 63 (60.00) | |
|     Non-myeloablative | 239 (30.88) | 202 (30.19) | 37 (35.24) | |
|     Reduced Intensity | 38 (4.91) | 33 (4.93) | 5 (4.76) | |
| GvHD Prophylaxis, n (%) | | | | 0.202 |
|     CNI + MMF | 140 (18.09) | 120 (17.94) | 20 (19.05) | |
|     CNI + MTX | 367 (47.42) | 323 (48.28) | 44 (41.90) | |
|     Campath | 139 (17.96) | 118 (17.64) | 21 (20.00) | |
|     PT-Cy | 77 (9.95) | 61 (9.12) | 16 (15.24) | |
|     Other | 51 (6.59) | 47 (7.03) | 4 (3.81) | |
| HCT-CI (median, IQR) | 3.00 (2.00–4.00) | 3.00 (2.00–4.00) | 3.00 (3.00–5.00) | 0.010 |
| Pre-transplant weight in lbs (median, IQR) | 185.00 (157.00–212.00) | 184.00 (157.00–211.00) | 188.00 (157.00–225.00) | 0.281 |
| Pre-Transplant BMI in kg/m2 (median, IQR) | 27.80 (24.45–31.64) | 27.79 (24.34–31.53) | 28.55 (24.75–32.79) | 0.172 |
| Pre-Transplant Albumin in g/dL (median, IQR) | 3.90 (3.60–4.20) | 3.90 (3.60–4.20) | 3.80 (3.50–4.00) | 0.042 |

**Table 2  Outcomes of patients who were not on a BB (No BB) or who were on a BB (Yes BB) peri-transplant (>4 days both before and after) to allo-HCT.**

| Parameter Statistic | All Patients N = 774 (100%) | No BB N = 669 (86%) | Yes BB N = 105 (14%) | P-Value |
|---|---|---|---|---|
| Acute GVHD time days (median, IQR) | 53.89 (28.94–211.58) | 53.89 (28.94–203.60) | 63.87 (33.93–352.31) | 0.187 |
| Acute GVHD occurrence, n (%) | | | | 0.036 |
| 0 | 276 (35.66) | 229 (34.23) | 47 (44.76) | |
| 1 | 498 (64.34) | 440 (65.77) | 58 (55.24) | |
| Acute GVHD grade (None vs Low vs High), n (%) | | | | 0.110 |
| No GvHD | 276 (35.66) | 229 (34.23) | 47 (44.76) | |
| 1 | 107 (13.82) | 95 (14.20) | 12 (11.43) | |
| 2+ | 391 (50.52) | 345 (51.57) | 46 (43.81) | |
| Acute GVHD grade (None/Low vs High), n (%) | | | | 0.139 |
| 0–1 | 383 (49.48) | 324 (48.43) | 59 (56.19) | |
| 2+ | 391 (50.52) | 345 (51.57) | 46 (43.81) | |
| Chronic GVHD time days (median, IQR) | 244.02 (136.73–468.08) | 245.52 (138.73–478.06) | 230.55 (123.76–403.21) | 0.322 |
| Chronic GVHD occurrence, n (%) | | | | 0.225 |
| 0 | 433 (55.94) | 380 (56.80) | 53 (50.48) | |
| 1 | 341 (44.06) | 289 (43.20) | 52 (49.52) | |
| Chronic GVHD grade, n (%) | | | | 0.373 |
| No GvHD | 433 (55.94) | 380 (56.80) | 53 (50.48) | |
| 1 | 81 (10.47) | 68 (10.16) | 13 (12.38) | |
| 2 | 260 (33.59) | 221 (33.03) | 39 (37.14) | |
| Length Of stay in days (median, IQR) | 87.83 (75.85–97.81) | 86.83 (75.85–96.81) | 89.82 (76.85–102.80) | 0.373 |
| Follow-up time in months (median, IQR) | 14.00 (6.00–47.00) | 14.00 (6.00–47.00) | 16.00 (6.00–47.00) | 0.602 |
| Post-transplant weight in lbs (median, IQR) | 170.00 (144.00–197.00) | 170.00 (144.00–196.00) | 169.00 (143.00–207.00) | 0.753 |
| Post-transplant BMI in kg/m2 (median, IQR) | 25.69 (22.57–29.10) | 25.61 (22.53–29.07) | 25.95 (22.74–30.27) | 0.437 |
| Post-transplant albumin in g/dL (median, IQR) | 3.80 (3.50–4.20) | 3.90 (3.50–4.20) | 3.75 (3.30–4.00) | 0.014 |
| Change in albumin (post-pre) (median, IQR) | 0.00 (−0.40–0.30) | 0.00 (−0.40–0.30) | −0.10 (−0.50–0.15) | 0.102 |

at D+90 was excluded from MVA to avoid *post-hoc* confounding since aGVHD commonly occurs before D+90. None were statistically significant (Table 3).

Additionally, we subdivided the Yes BB group by BB Mechanism (Table 4). Use of a non-selective BB (propranolol, sotalol, nadolol) was associated with lower pre-transplant weight compared to vasodilating BB (carvedilol, labetalol) and selective BB (atenolol, metoprolol, nebivolol) (136 lbs *vs.* 165 lbs *vs.* 174 lbs, respectively, $p = 0.034$). The use of vasodilating BB trended towards reduced LOS, however this just missed the cut-off for statistical significance ($p = 0.054$). There were no other significant differences in either primary or secondary outcomes for the mechanism-based subgroups (all $p > 0.05$) (Table 5).

## DISCUSSION

Our study found an association with decreased incidence of aGVHD in allo-HCT patients that had peri-transplant BB exposure on univariate analysis ($p = 0.036$); though this association did not remain significant on multivariate analysis ($p = 0.150$), suggesting
**Table 3  Multivariate analysis of covariates of aGVHD for allo-HCT patients.** Analysis includes the hazard ratio (HR).

| Covariates | Hazard ratio (HR) | Lower 95% CI | Upper 95% CI | *P*-value |
|---|---|---|---|---|
| Beta-blocker use | | | | |
| No BB | Reference | | | |
| Yes BB | 0.769 | 0.537 | 1.100 | 0.150 |
| Age at transplant | 1.003 | 0.994 | 1.012 | 0.466 |
| HCT-CI | 1.015 | 0.943 | 1.094 | 0.686 |
| Pre-Transplant Albumin | 1.075 | 0.813 | 1.422 | 0.610 |
| GVHD Prophylaxis | | | | |
| CNI + MTX | Reference | | | |
| CNI + MMF | 0.934 | 0.652 | 1.338 | 0.710 |
| Campath | 0.997 | 0.617 | 1.610 | 0.990 |
| PT-Cy | 0.760 | 0.530 | 1.087 | 0.133 |
| Other | 1.052 | 0.626 | 1.769 | 0.848 |

**Table 4  Breakdown of patients who were on a BB (Yes BB, *n* = 105) peri-transplant (>4 days both before and after) to allo-HCT by BB type and BB mechanism.**

| BB type (generic name) | BB mechanism | *n* (%) |
|---|---|---|
| Atenolol | Selective | 12 (11.4) |
| Carvedilol | Non-selective+alpha | 26 (24.8) |
| Metoprolol | Selective | 57 (54.3) |
| Propranolol | Non-selective | 5 (4.8) |
| Labetalol | Non-selective+alpha | 2 (1.9) |
| Sotalol | Non-selective | 1 (1.0) |
| Nadolol | Non-selective | 2 (1.0) |
| Nebivolol | Selective | 1 (1.0) |

that other factors may contribute to this trend, or that covariates may have overshadowed the BB effect. Therefore, we are hesitant to fully support a role for BBs as a prophylactic intervention to prevent aGVHD. One clinical study evaluated the risk of developing cGVHD for patients on a non-selective BB during the time of allo-HCT. While their results trended towards significance, that BBs were protective against cGVHD, they lacked sufficient power to detect a definitive difference. In our study, we found no association between BB and cGVHD occurrence ($p = 0.947$) or severity ($p = 0.988$). However, it is possible that a few days of peri-transplant BB exposure was insufficient to exert a lasting impact on this long-term outcome, given the median time to onset of cGVHD was 244 days (approximately 7.7 months) in our cohort. Future studies should include BB dosage and duration to better examine the impact of BB intake on long-term outcome evaluation of survival outcomes, including cGVHD.

The broader literature supports BB use as a potential adjunct in the care of cancer patients (*Jansen et al., 2014*; *Wang et al., 2013*; *Hwa et al., 2017*; *Heliste et al., 2022*; *Patel et al., 2023*; *Tan et al., 2019*). Additionally, previous studies have documented the role of b-AR

Bergens et al. (2025), *PeerJ*, DOI 10.7717/peerj.19822

**Table 5  Outcomes of patients who were on a BB peri-transplant (>4 days both before and after) to allo-HCT by BB mechanism.**

| Parameter Statistic | All Patients N = 105 (100%) | Non-selective N = 7 (6.67%) | Vasodilating N = 28 (26.67%) | Selective N = 70 (66.67%) | P-Value |
|---|---|---|---|---|---|
| Acute GVHD Time Days (median, IQR) | 63.87 (33.93–352.31) | 62.88 (14.97–211.58) | 68.37 (38.92–653.21) | 66.37 (28.94–352.31) | 0.746 |
| Acute GVHD Occurrence, n (%) | | | | | 0.258 |
| 0 | 47 (44.76) | 2 (28.57) | 16 (57.14) | 29 (41.43) | |
| 1 | 58 (55.24) | 5 (71.43) | 12 (42.86) | 41 (58.57) | |
| Acute GvHD Grade (None vs Low vs High), n (%) | | | | | 0.054 |
| No GvHD | 47 (44.76) | 2 (28.57) | 16 (57.14) | 29 (41.43) | |
| 1 | 12 (11.43) | 3 (42.86) | 2 (7.14) | 7 (10.00) | |
| 2+ | 46 (43.81) | 2 (28.57) | 10 (35.71) | 34 (48.57) | |
| Chronic GVHD Time Days (median, IQR) | 230.55 (123.76–403.21) | 276.46 (211.58–724.57) | 228.55 (86.83–369.27) | 218.57 (118.77–412.19) | 0.393 |
| Chronic GVHD occurrence, n (%) | | | | | 0.947 |
| 0 | 53 (50.48) | 3 (42.86) | 14 (50.00) | 36 (51.43) | |
| 1 | 52 (49.52) | 4 (57.14) | 14 (50.00) | 34 (48.57) | |
| Chronic GVHD grade, n (%) | | | | | 0.988 |
| No GvHD | 53 (50.48) | 3 (42.86) | 14 (50.00) | 36 (51.43) | |
| 1 | 13 (12.38) | 1 (14.29) | 3 (10.71) | 9 (12.86) | |
| 2+ | 39 (37.14) | 3 (42.86) | 11 (39.29) | 25 (35.71) | |
| Overall survival, n (%) | | | | | 0.766 |
| 0 | 46 (43.81) | 2 (28.57) | 13 (46.43) | 31 (44.29) | |
| 1 | 59 (56.19) | 5 (71.43) | 15 (53.57) | 39 (55.71) | |
| Non-relapse mortality, n (%) | | | | | 0.947 |
| 0 | 53 (50.48) | 3 (42.86) | 14 (50.00) | 36 (51.43) | |
| 1 | 52 (49.52) | 4 (57.14) | 14 (50.00) | 34 (48.57) | |
| Length of stay in days (median, IQR) | 89.82 (76.85–102.80) | 100.80 (81.84–109.78) | 84.83 (67.87–89.82) | 91.82 (76.85–111.78) | 0.054 |
| Follow-up time in months (median, IQR) | 16.00 (6.00–47.00) | 22.00 (10.00–91.00) | 21.00 (3.00–37.00) | 14.00 (6.00–51.00) | 0.589 |
| Post-transplant weight in lbs (median, IQR) | 169.00 (143.00–207.00) | 136.00 (134.00–139.00) | 165.00 (149.00–207.00) | 174.00 (149.00–211.50) | 0.034 |
| Post-transplant BMI in kg/m2 (median, IQR) | 25.95 (22.74–30.27) | 20.74 (20.14–25.08) | 26.81 (24.59–31.26) | 26.22 (22.84–31.15) | 0.118 |
| Post-transplant albumin in g/dL (median, IQR) | 3.75 (3.30–4.00) | 3.85 (3.50–4.20) | 3.80 (3.40–3.90) | 3.60 (3.20–4.10) | 0.754 |
| Change in albumin (post-pre) (median, IQR) | −0.10 (−0.50–0.15) | −0.05 (−0.10–0.60) | 0.00 (−0.50–0.10) | −0.20 (−0.60–0.20) | 0.403 |

signaling in influencing proinflammatory pathways and immune cell function (*Ferrara, Cooke & Teshima, 2003*; *Holler, 2002*; *Piper & Drobyski, 2019*; *Jankovic et al., 2013*; *Knight et al., 2020*; *Kuo et al., 2021*; *Heliste et al., 2022*; *Mohammadpour et al., 2018*; *Pasha, Calvani & Favre, 2021*; *Powell et al., 2013*; *Maestroni, 2020*; *Méndez-Ferrer, Battista & Frenette, 2010*; *Wang & Cao, 2019*). Recent laboratory studies in murine models have explored the b-AR role in GVHD modulation, specifically b2-AR signaling, in the modulation of GVHD from allo-HCT. A single study demonstrated that b2-AR activation ameliorated aGVHD, which opposes our hypothesis (*Mohammadpour et al., 2020*). The same team found that blockade of b2-AR improved GVT without impacting GVHD (*Mohammadpour et al., 2018*). Another team demonstrated that cold stress increased norepinephrine production, leading to excessive b-AR signaling and suppression of GVHD, which was reversed with b2-AR antagonists, supporting a role of b2-AR signaling in modulating GVHD severity (*Leigh et al., 2015*). Thus, there exists some data that there is potential impact in the allo-HCT population by modulating the adrenergic receptor; however, our results would not support the use of BB for survival benefits in allo-HCT. Possible explanations for the discrepancy between our findings and prior studies include differences in sample size, patient population, or BB dosing regimens. Additionally, variations in cGVHD diagnostic criteria and clinical management strategies across institutions may contribute to differing results. It is important to note that the lack of statistical significance in survival outcomes suggests neither benefit nor harm from BBs, consistent with current data on their pharmacological safety.

While BB use did not impact post-HCT weight overall ($p = 0.753$), subgroup analysis revealed that patients receiving non-selective BBs were associated with lower post-HCT weight compared to vasodilating and selective BBs ($p = 0.034$). The use of vasodilating BB trended towards reduced LOS ($p = 0.054$). Vasodilating BBs have been shown to reduce LOS in heart failure patients and those undergoing cardiothoracic surgery, likely due to their effects on hemodynamic stability and adrenergic modulation (*Butler et al., 2006*; *Coleman et al., 2004*; *Fowler et al., 2001*; *Packer et al., 2002*). While this relationship between BBs and LOS has not been explored in all HCT patients, our preliminary findings from our subgroup analysis suggest that BB types might variably impact a patient's weight, and possibly LOS, during the peri-HCT period. Although further studies would need to substantiate these results and elucidate the underlying mechanism.

Several limitations of our study must be acknowledged. Our study is limited by the retrospective nature of the data collection. There are also underlying differences between the Yes BB and No BB groups, especially since the reasoning for being on a BB prior to allo-HCT is generally a co-morbidity that would increase HCT-CI and the potential risk of adverse events. While balanced in terms of demographics (except for HCT-CI), the Yes BB and No BB groups were unbalanced in terms of sample size, which could have contributed to the lack of statistical power to see a difference in additional outcome measures. While we attempted to control for confounding variables through multivariate analysis, residual confounding cannot be entirely excluded. Another limitation is the change in the use of cyclophosphamide during our cohort, where GVHD outcomes drastically improved, which could diminish the overall power we are able to detect. Lastly, the limited sample sizes in

our subgroup analysis reduced the statistical power to detect differences, underscoring the need for larger prospective or randomized studies to strength any inference.

In conclusion, our study suggests that BB use does not significantly affect outcomes in allo-HCT patients, including aGVHD; however, subgroup analysis suggest that BB type may differentially affect post-HCT weight. While our study demonstrates that BBs remain pharmacologically safe in this population, our results do not support their routine use for modification of allo-HCT outcomes. Future prospective studies are necessary to substantiate these preliminary findings and explore the role of BBs in peri-HCT management to better understand their clinical implications and therapeutic potential.

### Funding
The authors received no funding for this work.

### Competing Interests
Anthony D. Sung's work has been funded by Merck, Novartis, Enterome, and Seres. He has received research product for studies from DSM/iHealth, Clasado, and BlueSpark Technologies. He has consulted for Targazyme, Acrotech, Geron, and Janssen. None of the parties mentioned provided funding for this study.

### Author Contributions

- Matthew A. Bergens conceived and designed the experiments, analyzed the data, prepared figures and/or tables, authored or reviewed drafts of the article, and approved the final draft.
- John T. Bokman conceived and designed the experiments, analyzed the data, prepared figures and/or tables, authored or reviewed drafts of the article, and approved the final draft.
- Ernaya J. Johnson conceived and designed the experiments, analyzed the data, prepared figures and/or tables, authored or reviewed drafts of the article, and approved the final draft.
- Matthew L. Braun conceived and designed the experiments, analyzed the data, prepared figures and/or tables, authored or reviewed drafts of the article, and approved the final draft.
- Yan Li conceived and designed the experiments, analyzed the data, prepared figures and/or tables, authored or reviewed drafts of the article, and approved the final draft.
- Amy T. Bush conceived and designed the experiments, analyzed the data, prepared figures and/or tables, authored or reviewed drafts of the article, and approved the final draft.
- Lauren Hill conceived and designed the experiments, authored or reviewed drafts of the article, and approved the final draft.
- Jolien Van Opstal conceived and designed the experiments, authored or reviewed drafts of the article, and approved the final draft.

- Alessandro Racioppi conceived and designed the experiments, authored or reviewed drafts of the article, and approved the final draft.
- Rebecca Fan analyzed the data, authored or reviewed drafts of the article, and approved the final draft.
- Sejal Kaushik analyzed the data, authored or reviewed drafts of the article, and approved the final draft.
- Edwin Alyea performed the experiments, authored or reviewed drafts of the article, and approved the final draft.
- Nelson Chao performed the experiments, authored or reviewed drafts of the article, and approved the final draft.
- Taewoong Choi performed the experiments, authored or reviewed drafts of the article, and approved the final draft.
- Cristina Gasparetto performed the experiments, authored or reviewed drafts of the article, and approved the final draft.
- Mitchell Horwitz performed the experiments, authored or reviewed drafts of the article, and approved the final draft.
- Richard Lopez performed the experiments, authored or reviewed drafts of the article, and approved the final draft.
- Sendhilnathan Ramalingam performed the experiments, authored or reviewed drafts of the article, and approved the final draft.
- Keith Sullivan performed the experiments, authored or reviewed drafts of the article, and approved the final draft.
- Paul Wischmeyer conceived and designed the experiments, analyzed the data, prepared figures and/or tables, authored or reviewed drafts of the article, and approved the final draft.
- Anthony D. Sung conceived and designed the experiments, performed the experiments, analyzed the data, prepared figures and/or tables, authored or reviewed drafts of the article, and approved the final draft.

## Human Ethics

The following information was supplied relating to ethical approvals (i.e., approving body and any reference numbers):

Approval for exemption was obtained from the Duke University institutional review board (Pro00103818).

## Data Availability

The raw data is available in the Supplementary File.

## Supplemental Information

Supplemental information for this article can be found online at http://dx.doi.org/10.7717/peerj.19822#supplemental-information.

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
