# Peer review of "A retrospective study on beta-blocker use and outcomes in hematopoietic stem cell transplant patients"

_PeerJ, doi:10.7717/peerj.19822_

## Round 0.1 · original submission · Major Revisions

Both reviewers have several important comments. Please respond to them in detail.

Reviewer 1 ·

Basic reporting

Failures:

The discussion section lacks clarity in certain sentences, such as, "Although we did not see an impact of BB as a whole on LOS (p=0.903), our subgroup analysis revealed...," which makes comprehension challenging.
Figure captions are not sufficiently detailed for standalone understanding.
Some of the references in the manuscript are not clearly connected to the study’s main ideas. For example, the authors mention previous research on beta-blockers improving outcomes in critical illness and cancer but don’t fully explain how those findings relate to reducing GVHD in stem cell transplants. "Beta-blockers (BB) are pharmacologically safe medications that are used clinically for cardioprotection by.........".Making these connections clearer would help the readers understand why this study is important.

Suggested Improvements:

Rewrite unclear sentences in the discussion to ensure they are concise and precise. For example, specify the clinical relevance of the observed trends.
Provide more detailed figure captions to make figures understandable without referring back to the main text.
Expand the introduction to better situate this study in the context of previous research on beta-blockers and hematopoietic stem cell transplantation.

Experimental design

Failures:

The study's retrospective design limits its ability to establish causation.
The criteria for selecting patients on beta-blockers (>4 days before transplant) may introduce bias and is not justified in sufficient detail.
Limited subgroup sample sizes (e.g., vasodilating BB group, n=33) reduce the power to detect significant effects.

Suggested Improvements:

Justify the use of retrospective design and acknowledge its limitations more thoroughly in the Methods section.
Provide additional details on how patients were selected for the beta-blocker group and why the >4-day threshold was chosen (clearly mention the inclusion and exclusion criteria to select patients)
Highlight the need for future prospective or randomized studies in the discussion to strengthen causal inference.

Validity of the findings

Failures:

Multivariate analysis did not yield statistically significant results, weakening the conclusions.
The findings on vasodilating beta-blockers and reduced length of stay are preliminary but are presented with limited mechanistic explanation or clinical relevance.
The manuscript lacks a thorough discussion on why chronic GVHD and survival outcomes did not differ across groups and also the effect of BB on different diseases that do allotransplant.

Suggested Improvements:

Provide a more nuanced interpretation of the lack of significance in multivariate analysis. For instance, consider whether covariates like HCT-CI may have overshadowed the beta-blocker effect.
Discuss the potential mechanisms through which vasodilating beta-blockers might reduce hospital stays, citing additional relevant literature.
Add a brief explanation of why chronic GVHD and survival outcomes might not have been affected by beta-blocker use.

Additional comments

Expand the discussion to address clinical implications, including whether vasodilating beta-blockers could be considered for future trials in transplant patients.
Strengthen the conclusion by clearly outlining the key takeaways while emphasizing the preliminary nature of the findings.

Reviewer 2 ·

Basic reporting

The authors conducted a meaningful retrospective study to evaluate the impact of BB usage in allo transplant patients. The article is very well written to provide sufficient background, methods, results and discussion. Raw dataset is shared to allow reviewers validate the results.

Experimental design

The study did not include the duration of BB usage in data analysis. BB usage was defined as at least 4 days within 100 days before allo-HSCT. There's no information on the duration of documented BB usage with 100 days prior to and after allo-HSCT. The survival outcome, acute and chronic GVDH were compared between groups of BB usage for a short duration before allo-HSCT. Post-HSCT BB status and usage duration may also affect survival outcome and GVHD, which was overlooked in this study. Chronic GVHD may occur several months after allo-HSCT. A short duration of BB usage before transplant may not have sufficient impact on the longer-term outcome months or years following allo-HSCT. Therefore, the study result of no benefit of BB on overall survival needs to be re-examined to include the post-HSCT BB usage.

I recommend the authors to:
1. Include information of documented BB usage and duration within 100 days pre- and post-HSCT to evaluate BB effect on acute GVHD.
2. Include duration of BB usage in the multivariant analysis to examine the BB effect on acute GVHD.
3. For chronic GVHD evaluation, include the duration of BB usage post-HSCT and evaluate if longer BB uses has any effect on chronic GVHD. A few days temporarily use of BB may not have any impact on long-term outcome after transplant. A previous study (reference 22) used >3 months for BB usage group.
4. Please report the median time to develop aGVHD and cGVHD in the results.
5. I reviewed the raw dataset of BB usage groups. The authors recorded BB use at anytime, BB before and after HSCT. I regrouped the BB usage to include those who had BB both before and after HSCT and defined it as "peri-HSCT BB group" to compare OS. 105 patients are in the peri-HSCT BB group and 669 patients in the no-BB group. Survival curve separates after 5-6 years between two peri-HSCT BB usage groups. 6-year survival rate was 51% (BB group) vs 35% (no BB group), p=0.03. Please see enclosed PDF of the OS curve. The incidence of acute GVHD remained significantly lower in peri-HSCT BB group than non-BB group (66% vs 55%, p<0.05). I think the message here is the duration of BB intake is an important factor to predict survival outcomes. The authors need to include the duration of BB use peri-HSCT to re-evaluate survival outcomes.

Validity of the findings

The survival results were labeled as "time from diagnosis", which is incorrect. In reviewing the raw dataset, the follow up days were calculated from the date of transplant. Please correct all information in the figures and text to indicate survival outcome evaluation to "time from HSCT".

Additional comments

A short duration of BB use pre-HSCT (> 4 days) is not sufficient to evaluate the impact on long-term survival outcome and chronic GVHD post-allo. The conclusion that BB use has no benefits on survival outcome may not be true due to inappropriate data analysis design. I have made several recommendations to re-examine the data and to include the information of duration of BB usage in the multivariant analysis. I would not recommend publish this study until the authors re-analyze the data to examine the effects of beta blockers use during peri-HSCT and duration of BB usage.

Annotated reviews are not available for download in order to protect the identity of reviewers who chose to remain anonymous.

---

## Round 0.2 · Minor Revisions

Reviewer 2 ·

Basic reporting

Authors have edited the manuscript according to the reviewer's comments.

Experimental design

The definition of BB usage was defined as > 4 days peri-transplant in this study. It would be more supportive if the authors could reference whether any previous studies have used > 4 days to categorize BB intake.

The median time to cGVHD in this study was 230. 55 (about 7.7 months). A short duration of BB intake for a few days is not long enough to examine its impact on long-term outcomes of cGVHD and overall survival. The study concluded that no significant difference in these outcomes may not be conclusive. I recommend that the authors add this limitation to the discussion as a possible explanation of why the finding of the BB impact on OS was different from previous studies.

Validity of the findings

Recommend authors to add a sentence in the limitation discussion to address the comment in #2. Suggest in future studies to include BB dosage and duration to better examine the BB intake on long-term outcome evaluation of survival and chronic GVHD.

Additional comments

none

---

## Round 0.3 · accepted · Accept

Thank you for making the requested changes to the manuscript.

Reviewer 2 ·

Basic reporting

The manuscript is well-written to meet the requirements for publication.

Experimental design

The authors have addressed all the reviewing comments and edited the manuscript accordingly.

Validity of the findings

I have reviewed the latest submission and don't have additional comments for editing.